# Variability in Motor Threshold during Transcranial Magnetic Stimulation Treatment for Depression: Neurophysiological Implications

**DOI:** 10.3390/brainsci13091246

**Published:** 2023-08-26

**Authors:** Alexis Bourla, Stéphane Mouchabac, Léonard Lorimy, Bertrand Crette, Bruno Millet, Florian Ferreri

**Affiliations:** 1Department of Psychiatry, Hôpital Saint-Antoine, AP-HP, Sorbonne Université, 75012 Paris, Franceflorian.ferreri@aphp.fr (F.F.); 2ICRIN Psychiatry (Infrastructure of Clinical Research In Neurosciences-Psychiatry), Brain Institute (ICM), Sorbonne Université, INSERM, CNRS, 75013 Paris, France; 3Clariane, Medical Strategy and Innovation Department, 75008 Paris, France; 4NeuroStim Psychiatry Practice, 75005 Paris, France; 5Institut du Cerveau, Service de Psychiatrie Adulte de la Pitié-Salpêtrière, AP-HP, Sorbonne Université, ICM, 75013 Paris, France

**Keywords:** treatment-resistant depression, transcranial magnetic stimulation, motor threshold

## Abstract

The measurement of the motor threshold (MT) is an important element in determining stimulation intensity during Transcranial Magnetic Stimulation treatment (rTMS). The current recommendations propose its realization at least once a week. The variability in this motor threshold is an important factor to consider as it could translate certain neurophysiological specificities. We conducted a retrospective naturalistic study on data from 30 patients treated for treatment-resistant depression in an rTMS-specialized center. For each patient, weekly motor-evoked potential (MEP) was performed and several clinical elements were collected as part of our clinical interviews. Regarding response to treatment (Patient Health Questionnaire-9 (PHQ-9) before and after treatment), there was a mean difference of −8.88 (−21 to 0) in PHQ9 in the Theta Burst group, of −9.00 (−18 to −1) in the High-Frequency (10 Hz) group, and of −4.66 (−10 to +2) in the Low-Frequency (1 Hz) group. The mean improvement in depressive symptoms was 47% (*p* < 0.001, effect-size: 1.60). The motor threshold changed over the course of the treatment, with a minimum individual range of 1 point and a maximum of 19 points (total subset), and a greater concentration in the remission group (4 to 10) than in the other groups (3 to 10 in the response group, 1 to 8 in the partial response group, 3 to 19 in the stagnation group). We also note that the difference between MT at week 1 and week 6 was statistically significant only in the remission group, with a different evolutionary profile showing an upward trend in MT. Our findings suggest a potential predictive value of MT changes during treatment, particularly an increase in MT in patients who achieve remission and a distinct “break” in MT around the 4th week, which could predict nonresponse.

## 1. Introduction

Major depressive disorder (MDD) is a prevalent psychiatric illness, with over 300 million people worldwide grappling with its debilitating effects, making it responsible for 10% of all disability-adjusted life years globally [1]. This condition is characterized by pervasive low mood, diminished interest in activities, cognitive impairment, and a range of somatic symptoms that severely hinder the quality of life of those affected [2]. Regrettably, a substantial proportion of patients fail to find relief from their symptoms with standard pharmacological treatments, such as selective serotonin reuptake inhibitors (SSRIs) or serotonin–norepinephrine reuptake inhibitors (SNRIs) [2]. This lack of response, often classified as treatment-resistant depression (TRD), is a major challenge in the field of psychiatry and poses a significant barrier to patient recovery. In search of more effective strategies to combat TRD, researchers have turned to neuromodulatory techniques like repetitive transcranial magnetic stimulation (rTMS). This non-invasive approach or NIBS (non-invasive brain stimulation), which involves stimulating the dorsolateral prefrontal cortex (DLPFC), has demonstrated promising results as a therapeutic modality for TRD [3]. Not only has rTMS been shown to elicit an antidepressant response in many patients, but the effects have also been found to be enduring, providing long-lasting relief for many individuals who respond positively to this treatment [4]. This sustained response represents a major advancement in the field, as achieving lasting effects is one of the central challenges in treating MDD.

However, not all patients with TRD respond to rTMS, and understanding the factors that influence responsiveness is an area of active investigation. Patient-related factors, such as individual variations in brain anatomy, neural circuitry, and the nature of the depressive symptoms (different biotypes linked to specific networks), may influence the degree of response. Concurrent psychiatric and medical conditions could also modulate the effectiveness of the treatment [5]. Moreover, treatment-related factors also play a role in determining rTMS efficacy. The specific location of stimulus application, particularly its precision and consistency in targeting the DLPFC, is believed to be critical for achieving optimal neuromodulatory and clinical effects [6]. Inconsistent or misplaced stimulation may fail to adequately modulate the desired neural circuits, thereby diminishing the antidepressant effect of rTMS.

An additional crucial treatment-related factor is the dose of the treatment, which is typically personalized according to the patient’s resting motor threshold (MT or rMT) [7]. This involves adjusting the intensity of the magnetic pulses based on the minimum amount of energy needed to elicit a motor response in a specific muscle group. The proper determination of MT is critical for ensuring that the delivered dose of rTMS is both safe and effective. It is worth noting that MT can vary over time due to a multitude of factors, including changes in the patient’s physiological state and the use of concurrent medications. Consequently, the routine reassessment of MT might be necessary for maintaining optimal rTMS treatment parameters throughout the course of the therapy [8].

A recent study looked at motor threshold (MT) data from 374 patients who received transcranial magnetic stimulation (TMS) for depression from 2000 to 2019 [8]. MT was measured on each day of treatment using the visual method. The study found that MT varied by an average of 5% from day to day. In some cases, MT varied by more than 25%. The researchers concluded that measuring MT only once at the start of treatment could lead to underdosing or overdosing, which could have negative effects on the treatment, and they suggested that MT should be measured daily or at least weekly. The recommended stimulation intensity range for TMS is 110–120% MT [9]. However, studies have shown that successful treatment can be achieved with stimulation intensities as low as 80%, 90%, or 100% MT [10]. This implies that it is unlikely that the variation in MT would be so great that the stimulation provided would be too low to be effective, and there is no evidence to suggest that a person’s MT and seizure threshold are related. This means that a decrease in MT does not necessarily mean an increase in the risk of seizure [11]. While it is possible that MT may vary from day to day during TMS treatment, the risk of inadequate treatment or increased risk of seizure due to this variation is not substantiated. Even though the daily re-determination of MT could be expensive and unjustified, a detailed assessment of the motor threshold (MT) remains necessary, as we hypothesize that MT translates interesting neurophysiological characteristics for assessing the potential response to treatment [12].

Using retrospective data from a TMS clinical center where MT is acquired and recorded every week along with several other clinical data, our main objective was to assess the variability of MT during treatment to study whether this could have an impact on the response to treatment.

## 2. Methods

We undertook a naturalistic retrospective study using data from patients treated at the NeuroStim Psychiatry Practice (NPP) in Paris (France) between September 2022 and May 2023, in accordance with the reference methodology MR-004 [13] established by the Commission nationale de l’informatique et des libertés (CNIL), which governs the processing of personal data for the purposes of study in France, particularly studies involving the reuse of data from healthcare. Patients were informed individually (consent) and collectively (posting), a CNIL declaration (n° 2230025v0) was completed, and a repository on the Health Data Hub was made.

### 2.1. Study Group

Eligible participants included adult patients who received DLPFC rTMS treatment for treatment-resistant depression using a DuoMag device (Deymed, CE marked n° 2275). Treatment-resistant depression (TRD) was categorized as any major depressive episode with the failure of 2 antidepressants, in the context of either major depressive disorder (MDD) or bipolar disorder, as per the Diagnostic and Statistical Manual of Mental Disorders [14]. A cut-off score of 10 on the Patient Health Questionnaire (PHQ9) scale was chosen to allow for inclusion (moderate severity). Only the data of patients with full treatment (30 sessions) were included. 

### 2.2. Data Collection

Several clinical data, including gender; age; lateralization; weight; height; tobacco, alcohol, and caffeine consumption; physical activity; physical illnesses; quality of life (EuroQol Group EQ5D [15]); medication; baseline and follow-up severity scores—PHQ9 [16], Generalized Anxiety Disorder 7 (GAD7) [17]; Insomnia Severity Index ISI [18]; Beck Depression Inventory 2 BDI-II [19]—daily stimulation settings (device, coil, stimulation side, stimulation intensity, frequency, total number of pulses per session); and weekly MT, were extracted from the electronic clinical database whenever available or using a pseudonymized questionnaire hosted on the HDS servers (Health Data Hosting). 

For anxiety disorder, the diagnosis was made if anxiety symptoms were associated with a score greater than 10 on the Generalized Anxiety Disorder 7 (GAD-7), a standardized assessment tool used to measure and evaluate the severity of anxiety symptoms. GAD-7 consists of 7 questions that inquire about commonly observed anxiety symptoms such as nervousness, excessive worry, restlessness, and difficulty relaxing. Each question is rated on a response scale of 0 to 3, reflecting the frequency of experienced symptoms (0 indicating “not at all” and 3 indicating “nearly every day”). For post-traumatic stress disorder (PTSD) and obsessive–compulsive disorder (OCD), this was a declarative element. To potentially assess confounding factors on MT, we asked patients to report their consumption of tobacco, caffeine, and alcohol. Alcohol consumption was scored as 0 (no consumption), 1 (1 to 2 drinks per day), 2 (3 to 7 drinks per day), 3 (7 to 10 drinks per day), or 4 (more than 10 drinks per day); coffee consumption was scored as 0 (no consumption), 1 (fewer than 2 cups or equivalent per day), 2 (3 to 4 cups or equivalent per day), 3 (5 to 6 cups or equivalent per day), or 4 (more than 6 cups or equivalent per day); smoking was rated as 0 (no consumption), 1 (fewer than 5 cigarettes per day), 2 (5 to 10 cigarettes per day), 3 (11 to 15 cigarettes per day), 4 (16 to 20 cigarettes per day), 5 (20 to 30 cigarettes per day), or 6 (more than 30 cigarettes per day). 

The motor hotspot (M1) was determined using the 10–20 EEG method (and the visual method if the 10–20 method was not accurate), and MT was determined from M1 using the motor-evoked potential in compliance with the current French guidelines [20], which allows for much more precise determination than the visual method. TMS technicians (BC) trained in these methods used an EEG cap and the visual method by identifying the cranial vertex at the intersection between the nasion–inion mid-sagittal line and the inter-tragus line. The stimulating coil’s center was then positioned on a point 5 cm from the vertex on the inter-tragus line, marking the start for the functional localization of M1. At a low TMS output setting, stimulation pulses were delivered, and the contralateral hand was equipped with a signal capture electrode to perform a motor-evoked potential. The motor threshold was found by progressively increasing stimulation intensity until we had a signal corresponding to the definition of a response, i.e., the acquisition of a pulse between 50 and 150 µV (see Figure 1). The location consistently eliciting the most significant pulse was defined as the M1. 

The treatment target in the dorsolateral prefrontal cortex was then determined using the Beam F3 method in compliance with the latest recommendations [21]. The following three protocols, usually used in routine care, could be used: bilateral theta burst with the protocol of Li et al. [22] (1800 pulses iTBS on the left DLPFC, 1800 pulses cTBS on the right DLPFC, 80% MT), the classic high-frequency protocol at 10 Hz on the left DLPFC (3000 pulses, 110% MT), and the classic low-frequency protocol at 1 Hz on the right DLPFC (1500 pulses, 110% MT). A single rTMS technician carried out all the targeting under the supervision of the doctor present, as well as all the treatments, in order to limit any potential variability that might have arisen from different operators. 

### 2.3. Data Analysis

The data were analyzed at the ICRIN (Infrastructure of Clinical Research in Neurosciences—Psychiatry), Brain Institute (ICM), Paris. All data were analyzed using IBM SPSS Statistic 24 and Excel. The clinical response to rTMS was calculated as the percentage reduction in PHQ9 or BDI-II depression severity scores at the final measurement, relative to the baseline (PHQ9 or BDI-response). Normalized MT (nMT) was calculated in each patient as the ratio of the weekly MT relative to MT on the first day (nMW2–6 = MW2–6/MW1). The data for continuous measurements (age, number of sessions, MT, BDI-II, PHQ9, PHQ9 and BDI-response, nMT) are presented as the mean ± standard error of the mean (SEM) and were analyzed, as appropriate, using unpaired-sample *t*-tests, analysis of variance (ANOVA), and longitudinal mixed effects regression analyses. The binary outcomes (stimulation protocol) are presented as the percentage of patients (%). The mean improvement in depressive symptoms was calculated as the percentage reduction between week 1 and week 6 on the PHQ9. Cohen’s d was used to calculate the effect size using Wolfram. 

## 3. Results

The clinical and demographic characteristics of the retrospective sample of depressed patients treated with rTMS are presented in Table 1. 

Regarding the response to treatment (PHQ9 before and after treatment), there was a mean difference of −8.88 (−21 to 0) for PHQ9 in the Theta Burst group, of −9.00 (−18 to −1) in the High-Frequency (10 Hz) group, and of −4.66 (−10 to +2) in the Low-Frequency (1 Hz) group. The mean improvement in depressive symptoms was 47% (*p* < 0.001, effect-size: 1.60).

Considering the subgroups, the effect sizes differ slightly (1.77 in the moderate group, 1.97 in the moderately severe group, 1.24 in the severe group).

There was no significant change in weight or BMI during treatment.

The other results are listed in Table 2. 

In terms of lifestyle habits, the results show a non-significant reduction in alcohol and caffeine consumption, and stability in tobacco consumption (Table 3).

In total, 56.7% of patients were considered responders to treatment (improvement greater than 50% in the PHQ9 score), and remission was considered to be a score of less than 5 on the PHQ9 at week 6 (29.3% of patients) (see Table 4). 

The motor threshold changed over the course of the treatment, with a minimum individual range of 1 point and a maximum of 19 points (total subset), and a greater concentration in the remission group (4 to 10) than in the other groups (3 to 10 in the response group, 1 to 8 in the partial response group, 3 to 19 in the stagnation group). We also note that the difference between MT at week 1 and week 6 is statistically significant only in the remission group (Table 5), with a different evolutionary profile (Figure 2), showing an upward trend in MT.

Differences in MT variability over time were found according to responder status (see Figure 3), with a different evolutionary profile between patients with symptom improvement (even partial) and those in the stagnation group. 

## 4. Discussion

This study highlights several important points. Firstly, the percentages of response and remission were relatively similar to what is usually found in the literature and other real-world studies [23], with a 30% remission rate and a 30% response rate. The number of partial responders, on the other hand, seemed to be higher in our sample. 

The main finding was a significant difference in MT evolution in the remission group compared with the other groups, with a tendency for the threshold to increase, whereas in some studies, what seems predictive of remission is a progressive decrease in the threshold [8]. Another phenomenon worth noting is that in the stagnation group, there was a clear “break” in MT at the 4th week. We suggest that this break, if found in larger samples, could potentially be predictive of nonresponse to treatment, which would need to be monitored. 

Comparatively, another study conducted by a different team reported significant day-to-day MT variability, with potential clinical relevance in terms of both efficacy and safety. However, their results are consistent with prior findings of only minor MT change across days at a group level. 

Few studies have assessed the relationship between MT and side effects, or efficacy, and their results seems to be inconclusive.

Zarkowski et al. [24] aimed to investigate the impact of rTMS on the resting motor threshold. The study involved 50 medication-free patients being treated for major depression as part of a large, multisite study sponsored by the US National Institutes of Health (NIH). The researchers measured rMT at four different points, including before and after the double-blind phase, and then, weekly during the open phase. The treatment involved seventy-five 4 s trains (3000 pulses per session) 10 Hz stimulation applied over 37.5 min with the coil over the left dorsolateral prefrontal cortex (DLPFC) at 120% rMT. The results showed no significant change in rMT during at least two weeks of prefrontal rTMS treatment. The average within-subject coefficient of variation was 6.58%, and on average, the final rMT was 2.45% lower than the baseline rMT. However, five subjects had changes of approximately 20% from the baseline, which could cause dosing and safety issues if undetected. 

However, our study seems to be in line with that of Bajbouj M et al. [25], which examined the influence of ten daily sessions of left dorsolateral prefrontal rTMS on motor cortical excitability, as indicated by motor-evoked potentials elicited via transcranial magnetic stimulation in 30 patients. They found that compared to non-responders, responders (33%) exhibited changes suggesting a decrease in cortical excitability (i.e., an increase in MT). 

However, our results should be interpreted in light of our study’s limitations. This study was primarily supported by a naturalistic cohort, without the use of sham rTMS stimulation. This limitation would be more important if we were in a causality-seeking dynamic; however, our objective was to demonstrate a variation in MT. Therefore, MT variability could be attributed to natural neurophysiological changes, the cumulative effects of rTMS, or other factors. While the lack of a controlled environment may be seen as a limitation, we believe it also strengthens our findings, as it more closely reflects MT variability in the general patient population. We used a retrospective design and we focused on the most accurately reported data. However, the size of our cohort necessarily induces a lack of power, and we were unable to assess the impact of certain confounding factors such as tobacco, coffee, and alcohol, though these are data that we are trying to collect. 

Despite the limitations of our study design, we believe our results raise important questions regarding the efficacy of treatment based on the assessment of MT. These findings may inform future studies on the predictive potential of MT variations during the course of the treatment, and the following two elements should be studied in studies of greater power: the increase in MT in a group of patients in remission (and potentially even in response, as found in the study by Bajbouj et al.), and the “break” in MT in the middle of the treatment (around the 4th week), which, on the other hand, could be predictive of an absence of response. 

In conclusion, our study provides valuable insights into the role of motor threshold (MT) evolution in the efficacy of repetitive transcranial magnetic stimulation (rTMS) treatment for major depression. Our findings suggest a potential predictive value of MT changes during treatment, particularly an increase in MT in patients who achieve remission and a distinct “break” in MT around the 4th week, which could predict nonresponse. These observations, if confirmed in larger studies, could significantly enhance the individualization of rTMS treatment protocols. However, the complexity of MT’s role and the variability observed underscore the need for further research, particularly in controlled, prospective studies that can systematically account for potential confounding factors. Despite the limitations of our study, we believe our findings contribute to the growing body of knowledge on rTMS treatment for major depression and highlight the importance of ongoing MT assessment in optimizing treatment efficacy and safety.

## Figures and Tables

**Figure 1 brainsci-13-01246-f001:**
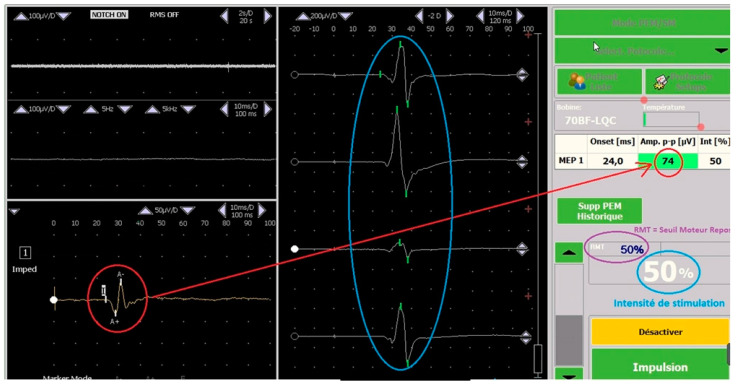
Motor-evoked potential. Red circle: peak-to-peak amplitude (µV), blue circle on the left: history of motor-evoked potential, purple circle: resting motor threshold target, blue circle on the right: stimulation intensity.

**Figure 2 brainsci-13-01246-f002:**
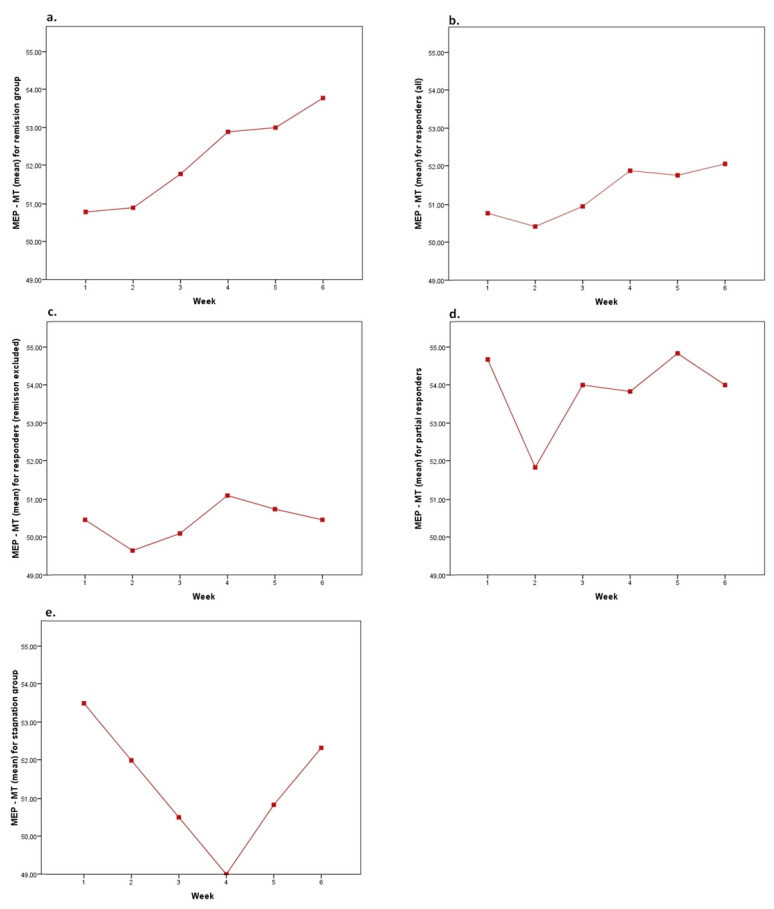
Evolution of MT over the 6-week treatment across different groups. (**a**) Mean MT for remission group, (**b**) Mean MT for responder group (including remitters), (**c**) Mean MT for responders group (excluding remitters), (**d**) Mean MT for partial responders group, (**e**) Mean MT for stagnation group.

**Figure 3 brainsci-13-01246-f003:**
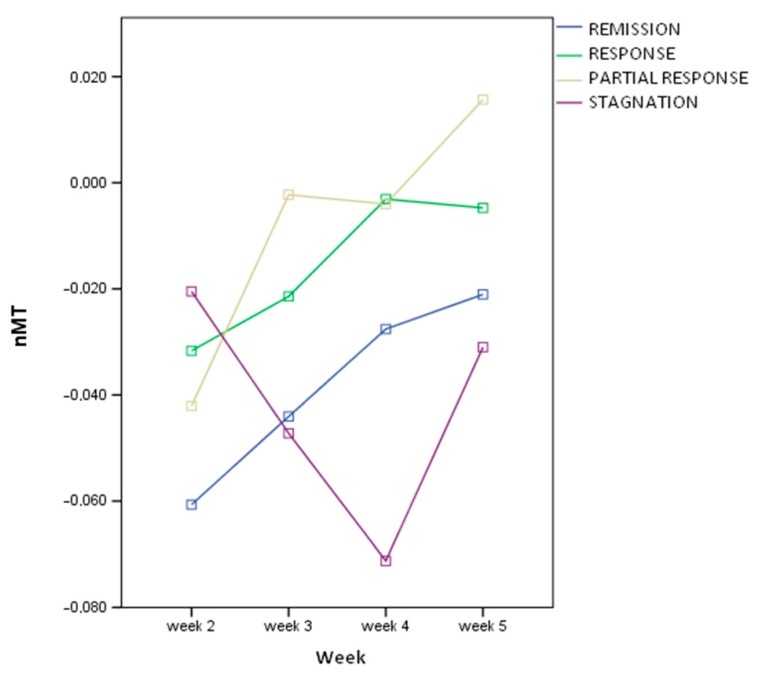
MT variability over time.

**Table 1 brainsci-13-01246-t001:** Clinical and demographic characteristics.

Total Sample (*n* = 30)	Characteristics
Age, mean (min, max)	44.2 (26–77)
Gender (% female)	66.7
Laterality (% right handed)	90.32
Education level	
-High School (%)	16.12
-Undergraduate (%)	45.17
-Graduate studies (%)	38.70
Family status	
-Single (%)	51.61
-Cohabitation (%)	19.35
-Married (%)	29.04
Physical illness	
-No other physical condition (%)	74.19
-Pulmonary disorder (COPD, SA) (%)	6.40
-Thyroid disorder (%)	3.22
-Fibromyalgia (%)	9.67
Depressive disorder	
-Unipolar depressive disorder (%)	83.88
-Bipolar depressive disorder (%)	16.12
-PHQ9 at baseline (mean, min, max)	17.17 (10–26)
Associated psychiatric condition	
-PTSD (%)	9.67
-Anxiety disorder (GAD > 10) (%)	58.06
Number of previous antidepressant trials (mean)	3
Protocol	
-HF (10 Hz) left DLPFC (%)	28.80
-Bilateral theta burst (%)	51.85
-LF (1 Hz) right DLPFC (%)	19.35
Total number of pulse over 6 weeks	
-In the HF 10 Hz group (mean)	87,900
-In the bilateral Theta Burst group (mean)	105,480
-In the LF 1 Hz group (mean)	45,000
Ongoing medication (% under)	
-Antidepressant alone	33.33
-Mood regulators alone **	13.33
-Antidepressant + mood regulator *	23.33
-Benzodiazepines in combination	40.00

* Including antipsychotics, ** including mood regulator combination (i.e., lamotrigine + aripiprazole).

**Table 2 brainsci-13-01246-t002:** Changes during treatment.

Results	Baseline	End of Treatment	*p* Value
	Mean (SEM)	Mean (SEM)	
PHQ9	16.83 (0.94)	8.77 (1.13)	0.016
-In Theta Burst subset	18.00 (1.35)	9.12 (1.79)	<0.001
-In High-Frequency (10 Hz) subset	17.43 (1.42)	8.43 (1.99)	0.004
-In Low-Frequency (1 Hz) subset	12.83 (1.51)	8.17 (1.47)	0.038
Depression severity (PHQ9)			
-Moderate (10–14)	12.00 (0.57)	5.80 (0.712)	<0.001
-Moderately severe (15–19)	17.00 (0.42)	8.90 (1.21)	<0.001
-Severe (>20)	22.50 (0.68)	11.60 (2.91)	0.003
Mean improvement for PHQ9			
-All groups	n.c	47.30 (5.39)	<0.001
-Moderate (10–14)	n.c	45.46 (8.10)	<0.001
-Moderately severe (15–19)	n.c	47.12 (7.56)	<0.001
-Severe (>20)	n.c	49.31 (12.5)	0.003
BDI-II	22.23 (2.96)	12.43 (2.55)	<0.001
GAD-7	12.37 (1.03)	7.43 (0.96)	0.001
EQ5D	18.30 (2.30)	20.23 (2.92)	0.007
ISI	11.57 (1.07)	6.80 (1.08)	<0.001
Weight	67.10 (2.36)	66.85 (2.47)	0.496
BMI	23.36 (0.66)	23.25 (0.66)	0.355

**Table 3 brainsci-13-01246-t003:** Variation in lifestyle habits.

Results	Baseline	End of Treatment	*p* Value
	Mean (SEM)	Mean (SEM)	
Alcohol	0.23 (0.14)	0.07 (0.04)	0.258
Tobacco	1.33 (0.33)	1.33 (0.33)	1.000
Caffeine	1.70 (0.16)	1.50 (0.17)	0.056

**Table 4 brainsci-13-01246-t004:** Response, partial response, stagnation, and worsening (PHQ9).

	*n*	%
Response	17	56.7
(Including remission)	(9)	(29.3)
Partial response	6	20.0
Stagnation	6	20.0
Worsening	1	3.3
Total	30	100

**Table 5 brainsci-13-01246-t005:** MT according to response profiles.

Results	Baseline	End of Treatment	*p* Value
	Mean (SEM)	Mean (SEM)	
MT (mean %)	52.35 (1.03)	52.33 (0.78)	0.705
-MT% in remission group	50.78 (1.42)	53.70 (1.49)	0.013
-MT% in all responders *	50.76 (0.94)	52.06 (1.07)	0.178
-MT% in exclusive responder **	50.45 (1.21)	50.45 (1.10)	1.000
-MT% in partial responders	54.67 (1.85)	54.00 (0.96)	0.650
-MT% in stagnation group	53.50 (4.03)	52.33 (2.31)	0.782

* Including remission, ** excluding remission.

## Data Availability

Data available on request.

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
