# Peer review of "Variability in Motor Threshold during Transcranial Magnetic Stimulation Treatment for Depression: Neurophysiological Implications"

_brainsci, 2023, doi:10.3390/brainsci13091246_

Round 1

Reviewer 1 Report

The motor threshold's measurement is crucial for determining stimulation intensity in rTMS treatment. Recommendations suggest weekly assessment due to threshold variability tied to neurophysiological traits. A study examined 30 depression patients in an rTMS center, tracking weekly MEP and clinical data. Response-wise, PHQ9 scores improved. Motor threshold shifted during treatment, indicating potential predictive value.

1. The meaning of the colored circles in Figure 1 is not clear. It recommends providing detailed information in the figure captions.

2. In Figure 2, it seems the line plots are from multiple groups. It will be nice to clearly indicate the error bar and the sample size.

Author Response

Dear Reviewer,

Thank you for your thoughtful feedback on our manuscript.

1. Regarding Figure 1, we apologize for any confusion caused by the colored circles in Figure 1. We realize that their significance was not adequately explained in the figure caption. We will revise the caption to provide a detailed description.

2. Each of our graphs represents data from a distinct group, and there is no direct comparison made between the groups in the graphical representation. Given this design choice, error bars wouldn't offer additional insight into inter-group differences or variabilities, as each group stands alone in its context. Furthermore, we have provided SEM values for several data point in a comprehensive table accompanying the graphs. Adding error bars to the graphs would replicate this information, potentially leading to redundancy. We believe that presenting the SEM in the table provides clarity without compromising the visual simplicity of the graphs.

We appreciate your understanding and valuable insights, which will undoubtedly enhance the clarity and comprehensiveness of our manuscript.

Warm regards,

Reviewer 2 Report

In abstract these terms have been mentioned without description: rTMS, MEP, PHQ9, MT

Introduction:

in lines 39-41: This condition is characterized by pervasive low mood, diminished interest in activities, cognitive impairment, and a range of somatic symptoms that severely hinder the quality of life of those affected. Add link to reference.

In lines 42-44: Regrettably, a substantial proportion of patients fail to find relief from their symptoms with standard pharmacological treatments, such as selective serotonin reuptake inhibitors (SSRIs) or serotonin-norepinephrine reuptake inhibitors. Why did you only mention these methods of treatment?

In lines 60-61: Concurrent psychiatric and medical conditions could also modulate the effectiveness of the treatment. Add link to references.

In lines 61-62: Moreover, treatment-related factors also play a role in determining rTMS efficacy.

Add link to references.

In lines 64-66: Inconsistent or misplaced stimulation may fail to adequately modulate the desired neural circuits, thereby diminishing the antidepressant effect of rTMS.

What do you mean by inconsistent or inadequate stimulation? Add link to reference.

 In line 71: delivered dose of rTMS is both safe and effective. Can you add some information about the complications of high doses? Add link to reference.

In lines 72-75: It's worth noting that the MT can vary over time due to a multitude of factors, including changes in the patient's physiological state and the use of concurrent medications. Consequently, routine reassessment of the MT might be necessary for maintaining optimal rTMS treatment parameters throughout the course of the therapy. Add link to reference

In lines 85-94:  Why did you discuss the risk of seizures in TMS treatment in detail?

In the introduction, you did not mention the pathophysiological effect of TMS in the treatment of major depressive disorder.

In lines 96-98: Can you be more specific about the purpose of the study?

Data collection

In line 133: PTSD and OCD, add description.

In line 158: in the dorsolateral prefrontal cortex (DLPFC). Delete description. The description was previously mentioned on line 49.

Where is the design of this study?

Add sample size determination.

Add inclusion criteria

Add exclusion criteria

What treatment are you comparing your study to?

Tell me, did all patients undergo pharmacotherapy? If yes, please describe which drug was used.

Can you explain the reason of MT changes during treatment?

Author Response

Dear Reviewer,

Thank you for taking the time to review our manuscript. We appreciate the constructive feedback and have addressed each of your concerns as outlined below:

We understand the need to clarify the mentioned terms (rTMS, MEP, PHQ9, MT) for a broader audience. We will expand these abbreviations at their first mention in the abstract.

Lines 39-41: We will add an appropriate reference to provide further context on the characteristics of the condition.

Lines 42-44: The mention of SSRIs and serotonin-norepinephrine reuptake inhibitors was to highlight commonly prescribed first line treatments.

Lines 60-61 & 61-62: While we understand the importance of referencing claims in scientific writing, the statements made in these lines are general observations based on clinical experience and are not specific claims derived from any particular study. Therefore, adding a direct reference might not be appropriate.

Lines 64-66: By "inconsistent or inadequate stimulation," we mean that if the rTMS stimulation is not applied with consistent parameters or if it is not targeted correctly, it may not affect the desired brain regions, thereby reducing its efficacy. This is a technical observation based on the principles of rTMS administration. However, if a specific reference is deemed necessary, we can look into it further.

Line 71: The statement about the delivered dose of rTMS being both safe and effective is a general claim. Typically, complications from high doses of rTMS include discomfort, headache, or the rare risk of seizure. The main point of this line was to highlight the overall safety and efficacy of the treatment, rather than to delve into potential complications. If a detailed discussion on complications is required, we could consider adding it.

Lines 72-75: Again, the observation here is based on general clinical knowledge and experience. Motor threshold (MT) variability due to physiological changes and medication use is a well-acknowledged aspect of rTMS administration. The appropriate reference is 8. Cotovio G, Oliveira-Maia AJ, Paul C, Faro Viana F, Rodrigues da Silva D, Seybert C, Stern AP, Pascual-Leone A, Press DZ. Day-to-day variability in motor threshold during rTMS treatment for depression: Clinical implications. Brain Stimul. 2021 Sep-Oct;14(5):1118-1125. doi: 10.1016/j.brs.2021.07.013. Epub 2021 Jul 27. PMID: 34329797. and it will be added to this line

Regarding the pathophysiological effect of TMS in the treatment of major depressive disorder, the main focus of our manuscript was not on the pathophysiological effects of TMS but on its clinical application. However, we appreciate your suggestion and will consider adding a brief overview of the pathophysiological mechanisms in the introduction even if it's not usually done in articles like ours, if it's the editor's wish. 

Lines 85-94: The risk of seizures is a crucial aspect of TMS treatment and is of significant concern to both clinicians and patients. It was discussed in detail to provide a comprehensive understanding of the treatment risks, even though such events are rare.

Lines 96-98: The purpose of the study is clearly stated at the end of the introduction section: "Using retrospective data from a TMS clinical center where MT is acquired and recorded every week along with several other clinical data, the main objective was to assess the variability of the MT during treatment to study whether this could have an impact on the response to treatment"

Line 133: We will expand the abbreviations PTSD and OCD at their first mention.

Line 158: Noted, we will delete the repeated description.

Design, sample size determination, inclusion, and exclusion criteria: As our study was a retrospective analysis of existing care data (MR004), traditional study design elements like sample size determination or inclusion/exclusion criteria aren't applicable in the same way as for a prospective study. This is described in the methodology. In this type of study, the inclusion criteria are described in the study group section.

Treatment comparison: Our study wasn't designed to compare with another treatment but to analyze patterns within the collected data.

Regarding pharmacotherapy: it is described in table 1.

Lastly, in response to the question about the reason for MT changes during treatment, our study aimed to elucidate patterns in MT changes rather than deduce causative factors. This was mentioned in our methodology section but we will add that in the discussion section. 

Round 2

Reviewer 2 Report

There is no doubt that the manuscript has been improved over the previous version. However, some unfinished tasks remain.

1.       In lines 91-95 . “This means that a decrease in MT does not necessarily mean an increase in the risk of seizure. In fact, studies have shown that the risk of seizure from TMS is very low. For example, one study found that the over- all rate of seizures from TMS was 0.31 per 10,000 treatments [11]”.

You have discussed in detail the risk of seizures in patients after MT. In fact, in many patients with seizures, the clinical manifestations of psychological disorders dominate without the characteristic manifestation of a seizure in the form of convulsions or absences. Indeed, MT may provoke epileptic seizures in patients with organic brain damage and in patients with epilepsy.

I do not think that this percentage of 0.3 per 10,000 is a rarity. In fact, if we exclude patients diagnosed with epilepsy and organic brain damage, this percentage will be much less.

I propose to exclude the discussion of the development of epilepsy in the introduction. Besides it is necessary to specify in inclusion/exclusion. that patients underwent EEG and MRI of the brain and were consulted by a neurologist before treatment. Otherwise, the treatment may be considered unreasonable or risky.

2.       Since we are talking about patients with a resistant form of depression, it is necessary to clarify the pharmacotherapy used in the previous treatment.

3.       The motor evoked potential is lower in post-stroke patients with changes in the motor cortex, in patients with multiple sclerosis, and in many other diseases. I'm sure you didn't include these patients in your study. Therefore, add this information to the exclusion criteria.

4.       In lines 91-92 : This means that a decrease in MT does not necessarily mean an increase in the risk of seizure.  Maybe meant that an increase  in MT does not necessarily mean an increase in the risk of seizure.  May be meant?

5.       Add method of sample size determination. You included 30 patients. You included 30 patients. How can we know if this number of patients is enough to get significant results. This is important for the relevance of your research. In this case, you must specify the primary endpoint (measurement) that allows you to calculate the minimum expected difference, the power value (in %) and the expected significance level (p-value). Many answers you can find in these sites:

https://clincalc.com/stats/samplesize.aspx

and

https://www.sealedenvelope.com/power/binary-superiority/

6.       Add design. Authors should inform readers about the total number of patients before randomization. It is also necessary to clarify the number and reasons for exclusion. After treatment, it is important to provide information about patients withdrawn from the study due to intolerance or side effects. It is desirable to use pictures and drawings.

Author Response

Dear Reviewer,

Thank you for the time and effort you've dedicated to reviewing our manuscript. We truly appreciate your insights and recommendations. However, it seems there may be some misunderstandings regarding the nature of our study, and we would like to clarify these points to ensure our work is evaluated in its appropriate context:

Discussion of the Development of Epilepsy: We concur with your suggestion and will omit the paragraph in question. Nevertheless, for your information, it's worth noting that in France, and indeed in most countries that routinely administer rTMS, an EEG or MRI is not mandatorily performed prior to an rTMS session. These evaluations are undertaken based on specific clinical indications.

Pharmacotherapy in Resistant Depression: Our definition of resistance has been clearly stipulated in the manuscript as resistance to two lines of treatment. As detailed in the table, we provide specifics regarding the number of previous treatments.

Medical history: We'd like to emphasize, as previously mentioned, that this is a retrospective study based on real-world data with the sole inclusion criterion being resistance to two lines of treatment. In studies of this nature, which are akin to real-world studies, no data is excluded. As detailed in our methodology, all patients with resistant depression were incorporated, regardless of age and medical history.

Lines 91-92 on MT: The context preceding this statement in our manuscript elucidates that it's improbable for the MT variation to be substantial enough to have an impact on the seizure threshold. There's no evidence suggesting a correlation between an individual's MT and their seizure threshold.

Sample Size Determination: In a retrospective study based on real-world care data, sample size determination is not performed a priori. Instead, the sample size is governed by the quantity of available data.

Study Design: As reiterated in our previous responses, it seems there's been a recurring misunderstanding. This is not a prospective randomized study. Instead, it's a retrospective study based on real-world care data.

We hope our clarifications help in presenting our study in its intended light and would be happy to provide any further explanations if required.

Warm regards,